# Influence of the Hypersensitivity to Low Dose Phenomenon on the Tumor Response to Hypofractionated Stereotactic Body Radiation Therapy

**DOI:** 10.3390/cancers15153979

**Published:** 2023-08-05

**Authors:** Eymeric Le Reun, Adeline Granzotto, Adeline Pêtre, Larry Bodgi, Guillaume Beldjoudi, Thomas Lacornerie, Véronique Vallet, Audrey Bouchet, Joëlle Al-Choboq, Michel Bourguignon, Juliette Thariat, Jean Bourhis, Eric Lartigau, Nicolas Foray

**Affiliations:** 1U1296 Unit, “Radiation: Defense, Health and Environment”, Centre Léon-Bérard, Inserm, 28 Rue Laennec, 69008 Lyon, France; eymeric.le-reun@inserm.fr (E.L.R.); adeline.granzotto@inserm.fr (A.G.); adeline.petre@lyon.unicancer.fr (A.P.); audrey.bouchet@inserm.fr (A.B.); joelle.al-choboq@inserm.fr (J.A.-C.); michel.bourguignon@inserm.fr (M.B.); 2Service de Radio-Oncologie, Centre Hospitalier Universitaire Vaudois (CHUV), 46 Rue du Bugnon, 1011 Lausanne, Switzerland; veronique.vallet@chuv.ch (V.V.); jean.bourhis@chuv.ch (J.B.); 3Département de Radiothérapie, Centre Léon-Bérard, 28 Rue Laennec, 69008 Lyon, France; guillaume.beldjoudi@lyon.unicancer.fr; 4Department of Radiation Oncology, American University of Beirut Medical Center, Riad El-Solh, Beirut 1107-2020, Lebanon; lb38@aub.edu.lb; 5Département de Radiothérapie, Centre Oscar-Lambret, 3 Rue Frédéric Combemale, 59000 Lille, France; t-lacornerie@o-lambret.fr (T.L.); e-lartigau@o-lambret.fr (E.L.); 6Département de Biophysique et Médecine Nucléaire, Université Paris Saclay, Versailles St. Quentin en Yvelines, 78035 Versailles, France; 7Département de Radiothérapie, Centre François-Baclesse, 3 Avenue du Général Harris, 14076 Caen, France; j.thariat@baclesse.unicancer.fr

**Keywords:** low-dose, radiation therapy, SBRT, LDRT, radiosensitivity, cancer, HRS phenomenon

## Abstract

**Simple Summary:**

We demonstrated the possible occurrence of the hypersensitivity to low dose (HRS) phenomenon in SBRT modality in both tumor and healthy cells. In HRS-positive cells, the response to SBRT was found exacerbated. Notably, a subset of highly damaged cells can appear and increase the efficiency of the treatment. Hence, each SBRT session can be viewed as hyperfractionated dose delivery by means of hundreds of low dose minibeams. To determine the HRS status of tumors and healthy tissues appears to be useful to increase SBRT efficiency and decrease the risk of adverse reactions.

**Abstract:**

Stereotactic body radiation therapy (SBRT) has made the hypofractionation of high doses delivered in a few sessions more acceptable. While the benefits of hypofractionated SBRT have been attributed to additional vascular, immune effects, or specific cell deaths, a radiobiological and mechanistic model is still needed. By considering each session of SBRT, the dose is divided into hundreds of minibeams delivering some fractions of Gy. In such a dose range, the hypersensitivity to low dose (HRS) phenomenon can occur. HRS produces a biological effect equivalent to that produced by a dose 5-to-10 times higher. To examine whether HRS could contribute to enhancing radiation effects under SBRT conditions, we exposed tumor cells of different HRS statuses to SBRT. Four human HRS-positive and two HRS-negative tumor cell lines were exposed to different dose delivery modes: a single dose of 0.2 Gy, 2 Gy, 10 × 0.2 Gy, and a single dose of 2 Gy using a non-coplanar isocentric minibeams irradiation mode were delivered. Anti-*γH2AX* immunofluorescence, assessing DNA double-strand breaks (DSB), was applied. In the HRS-positive cells, the DSB produced by 10 × 0.2 Gy and 2 Gy, delivered by tens of minibeams, appeared to be more severe, and they provided more highly damaged cells than in the HRS-negative cells, suggesting that more severe DSB are induced in the “SBRT modes” conditions when HRS occurs in tumor. Each SBRT session can be viewed as hyperfractionated dose delivery by means of hundreds of low dose minibeams. Under current SBRT conditions (i.e., low dose per minibeam and not using ultra-high dose-rate), the response of HRS-positive tumors to SBRT may be enhanced significantly. Interestingly, similar conclusions were reached with HRS-positive and HRS-negative untransformed fibroblast cell lines, suggesting that the HRS phenomenon may also impact the risk of post-RT tissue overreactions.

## 1. Introduction

Over the last decades, numerous technological advances in radiotherapy (RT)—notably, via stereotactic approaches—have permitted better tumor targeting and a drastic reduction in the volume of irradiated healthy tissues. This is, notably, the case of the stereotactic body radiation therapy (SBRT). Considering the great diversity of SBRT protocols, with various doses per fraction, number of fractions, dose gradient, etc., it appears more rigorous to describe SBRT from its physical features, i.e., as a RT modality delivering many non-coplanar minibeams converging to the tumor with sub-millimetric accuracy, as defined by the International Commission on Radiation Units and Measurements (ICRU) [1,2,3,4,5,6]. Particularly, hypofractionated SBRT has been shown to improve anti-tumor efficiency and decrease the volume of irradiated healthy tissues for numerous tumor indications, including spine metastasis [7,8,9], prostate cancer [10], liver metastasis [11], as well as lung carcinoma and metastasis [7,12,13].

In stereotactic approaches, a precise tumor target is reached in each session by hundreds of minibeams delivering subfractions of Gy [14]. For example, during one SBRT session, the targeted tumor may be exposed to a hyperfractionation of low doses of the order of cGy. Furthermore, some minibeams may overlap in the same regions of the targeted tumor; hence, in a given tumor region, a session of SBRT results in some repeated low doses that are separated by some seconds to some minutes (Figure 1).

Interestingly, at the ranges of doses and dose-rates involved during an SBRT session, the hypersensitivity to low dose (HRS) phenomenon can occur and result in enhancing the effect of the low dose to an effect that is equivalent to a dose 5 to 10 times greater [15,16,17]. Any HRS-positive tumor treated by SBRT could, therefore, elicit a considerable dose enhancement effect caused by HRS. (Figure 1). At high dose-rate (some Gy/min), the maximal HRS effect generally ranges from 0.1 to 0.8 Gy. However, the dose at which the maximal HRS effect occurs has been shown to decrease with dose-rate reaching a maximal HRS effect in the mGy to cGy range [18]. By using unrepaired DSB as an endpoint, such dose enhancement may be quantified after one single SBRT session.

The present study aims to investigate the potential influence of the HRS phenomenon in the response of tumor and normal cells exposed to SBRT, as defined above. In order to verify such a hypothesis, four HRS-negative and two HRS-positive human tumor cell lines were exposed to a single dose of 0.2 Gy, 2 Gy, 10 × 0.2 Gy, and a single dose of 2 Gy using a non-coplanar isocentric minibeams irradiation mode at Centre Léon-Bérard (CLB), Lyon, France and at Centre Hospitalier Universitaire Vaudois (CHUV), Lausanne, Switzerland. There were also one HRS-negative and one HRS-positive human untransformed fibroblast cell lines used to evaluate the influence of the HRS status on the response of healthy tissues to SBRT. The immunofluorescence against the phosphorylated forms of the variant X of the H2A histone (γH2AX), a current biomarker of DSB recognized by the non-homologous end-joining (NHEJ), the most predominant DSB repair pathway in mammalians [19,20], was applied to each condition to investigate the DSB repair and signaling response.

## 2. Materials and Methods

### 2.1. Cell Lines

There were four HRS-positive and two HRS-negative commercial human tumor cell lines used in this study. Their major biological and radiobiological features were detailed in Table 1. All the experiments with tumor cell lines were performed in the log phase of growth with similar relative distributions in the cell cycle phases (70–80% in G0/G1; 10–15% in S and 10–20% in G2/M) [21]. In addition, to evaluate the influence of the HRS status on the response of healthy tissues to SBRT, one HRS-negative (AG1521) and one HRS-positive (13HNG) human untransformed fibroblast cell line were irradiated in the same conditions as those used for tumor, to the notable exception that they were irradiated in plateau phase of growth [22]. While the AG1521 fibroblast cell line is provided from a commercial cell repository (Table 1), the 13HNG cell line belongs to the COPERNIC collection (N.F.‘s lab), which has been abundantly documented and composed of radioresistant and radiosensitive fibroblast cell lines derived from RT-treated patients. The COPERNIC radiobiological database is protected under the reference IDDN.FR.001.510017.000.D.P.2014.000.10300. All sampling protocols of the COPERNIC collection were approved by the national ethical committee in agreement with the current national regulations. The resulting cells were declared under the numbers DC2008-585, DC2011-1437, and DC2020-3957 to the Ministry of Research. All the cells used in this study were routinely cultured with Dulbecco’s modified Eagle’s minimum medium (DMEM) (Gibco-Invitrogen-France, Cergy-Pontoise, France), which was supplemented with 20% fetal calf serum, penicillin, and streptomycin.

### 2.2. Irradiation

Irradiations simulating a hyperfractionated RT were performed with a 6 MV X-rays medical irradiator (SL 15 Philips) (dose-rate: 4 Gy.min^−1^) at Centre Léon-Bérard (Lyon, France) [23]. In all the experiments, a dose of 2 Gy was chosen as reference since it simulates a current dose per RT session. Cells in Petri dishes were exposed in a static irradiation described elsewhere [23]. Dosimetry was certified by radiophysicists of the Centre Léon-Bérard. SBRT Irradiations were performed with a CyberKnife^TM^ treatment unit (Accuray Inc, Sunnyvale, CA, USA) at Centre Léon Bérard (Lyon, France) or at CHUV (Lausanne, Switzerland) [24]. First, cells in Petri dishes were exposed with static irradiation: dishes were positioned on the floor on 10 cm of water equivalent plates, for backscattering purposes, and under 5 cm of water equivalent plates. A fixed collimator of 60 mm was used. In these conditions, dose-rate to the cells was estimated to about 1.5 Gy.min^−1^. Cells were irradiated by considering three schemes: 2 Gy delivered in one time (“2 Gy”), 0.2 Gy delivered in one time (“0.2 Gy”), and 2 Gy delivered in 10 subfractions of 0.2 Gy, with a 2 min time interruption between each fraction, leading to a planned delivery time of about 20 min (“10 × 0.2 Gy”). A second setup was considered with a CyberKnife^TM^ treatment unit: Petri dishes were positioned inside a homemade phantom containing fiducials that could be seen by the CyberKnife^TM^ system, and a CT scan was performed. Multiplan^TM^ (Accuray Inc., Sunnyvale, CA, USA) treatment planning system was used to create a clinical plan, and the fixed 20 mm, 35 mm, and 50 mm collimators were chosen. A homogeneous dose of 2 Gy was planned for the target volume, containing the Petri dishes plus isotropic extension margins, of 5 mm to account for setup uncertainties. The treatment consisted of 82 minibeams, including 8, 71, and 3 minibeams provided by fixed 20, 35, and 50 mm collimators, respectively. The planned delivery time was about 20 min. It will be further referred to, in this paper, as the (“2 Gy Cyber”) condition.

### 2.3. Cell Survival

Standard clonogenic assays were used for the assessment of intrinsic radiosensitivity, as previously described [25,26]. For HRS-negative cells, the linear-quadratic (LQ) model describes the cell survival S as a function of dose D, as follows:S(D) = exp(−αD − βD^2^)(1)
in which α and β are adjustable parameters to be determined.

For HRS-positive cells, the induced repair model describes the cell survival S as a function of dose D, similar to the LQ model, but by integrating the HRS phenomenon with the following modifications on the α parameter, by defining the HRS variant LQ model as follows:S(D) = exp(−α_res_ (1 + g exp(−D/Dc))D − βD^2^)(2)
in which Dc is an adjustable parameter and g is the amount by which α, at the very low doses (α_sen_), is larger than α at high doses (α_res_). Hence, α_sen_ = α_res_ (1 + g) for the hyperradiosensitive cells [27] (Figure 1). The HRS/IRR ratio can also be used to quantify the extent of the HRS phenomenon. It is defined as the ratio of high and low doses, providing the lowest survival values at low doses and corresponding to the maximal HRS effect.

### 2.4. Immunofluorescence

The immunofluorescence protocol and foci scoring were described elsewhere [23]. Briefly, cells were fixed in 4% (*w/v*) paraformaldehyde solution for 10 min at room temperature, and they were permeabilized in 0.5% Triton X-100 solution for 5 min at 4 °C. Primary and secondary antibody incubations were performed for 40 and 20 min at 37 °C, respectively. Anti-*γH2AX^ser139^* antibody (#05-636; Merck Millipore, Burlington, VT, USA) was used at a 1:800 ratio. Anti-mouse fluorescein (FITC) secondary antibody was used at a 1:100 ratio. Slides were mounted and counterstained in 4′,6′ Diamidino-2-Phenyl-indole (DAPI)-stained Vectashield (Abcys, Paris, France), and nuclei were examined with a BX51 Olympus fluorescence microscope. For each of the three independent replicates, 100 nuclei were analyzed. The foci scoring procedure applied here has received the certification agreement of CE mark and ISO-13485 quality management system norms. Our foci scoring procedure also developed some features that are protected in the frame of the patents (FR3017625 A1 and FR3045071 A1).

### 2.5. Statistical Analysis

Statistical analysis was performed by using Kaleidagraph v4 (Synergy Software, Reading, PA, USA), Graphpad Prism (San Diego, CA, USA), and MATLAB R2020B (MathWork, Natick, MA, USA). Each experiment is the result of three independent replicates, so the mean is given with the standard error of the mean (SEM).

## 3. Results

### 3.1. HRS Status of the 6 Tumor Cell Lines Used in This Study

Using a standard orthovoltage irradiator, delivering X-rays at high dose-rate, clonogenic cell survival assays were applied to the HT29, A549, U373, U87, LN229, and SNU475 cell lines. With regard to cell survival, while the U373 and SNU475 cells were found to be HRS-negative by obeying the classical LQ model data fitting, the HT29, A549, U87, and LN229 cells showed a specific hyperradiosensitivity in the range of 0.1 to 0.5 Gy, and they obeyed the HRS variant LQ model (Figure 2; Table 2). It is noteworthy that the surviving fractions, at 2 Gy (SF2) of the HRS-positive LN229 and the HRS-negative U373 cells, were not significantly different (61 ± 3% and 58 ± 6%, respectively; *p* > 0.8), while the SF2 values of the LN229 (HRS-positive) and the A549 (HRS-positive) were significantly different (61 ± 3% and 80 ± 4%, respectively; *p* < 0.001). These findings suggest that the extent of HRS phenomenon may be observed independently of the SF2 value. A similar conclusion can be reached with the cell cycle since all the cell lines have been irradiated in similar distributions of cell cycle phases. Our results were found to be in agreement with literature data [28,29,30] (Figure 2).

### 3.2. γH2AX Foci Kinetics of the 6 Tumor Cell Lines Used in This Study

The formation of nuclear γH2AX foci reveals the DSB, which is recognized by the non-homologous end-joining pathway (NHEJ), the most predominant DSB repair and signaling in mammalians [20]. The early (10 min) γH2AX foci document the recognition step of the DSB managed by NHEJ, while the residual (24 h) γH2AX foci reflect the DSB repair step [20]. After 2 Gy, the anti-*γH2AX* immunofluorescence revealed a maximal number of γH2AX foci reached at 10 min post-irradiation and corresponding to 80 ± 5 and 78 ± 4 γH2AX foci in the U373 and the HT29 cells (*p* > 0.2), respectively. Conversely, in the A549 cells, this number was found to be significantly lower (65 ± 4 γH2AX foci; *p* < 0.005), suggesting an impaired DSB recognition, which was in agreement with literature [20] (Figure 3). At 24 h after 2 Gy, the amounts of residual γH2AX foci in the three cell lines tested were found to be similar (*p* > 0.2), suggesting that, at 2 Gy, (1) the DSB recognized by H2AX phosphorylation are repaired similarly, and (2) the differences between cell survival (SF2) values are likely due to DSB that are not recognized by NHEJ and not revealed by γH2AX foci [17] (Figure 3). After a single dose of 0.2 Gy, the number of γH2AX foci were about 10 times lower for all the cells. However, the HRS-positive cells elicited a lower number of γH2AX foci, assessed 10 min post-irradiation, than the HRS-negative cells (*p* < 0.05), suggesting a DSB recognition impairment at 0.2 Gy. Furthermore, the HRS-positive cells elicited a higher number of γH2AX foci 24 h post-irradiation than the HRS-negative cells (*p* > 0.001), suggesting a slower DSB repair (Figure 3). Altogether, these findings suggest that, at 0.2 Gy, unlike the HRS-negative U373 cells, the HRS-positive HT29 and A549 cells showed a lack of DSB recognition by NHEJ pathway and a larger amount of unrepaired DSB, which documents the HRS phenomenon (Figure 3). Similar conclusions were reached with the HRS-positive U87 and LN229 cells and the HRS-negative SNU475 cells.

Among the most current questions with regard the relevance of the HRS phenomenon is how a low dose, such as 0.2 Gy, can provide a similar lethal effect to higher doses such as 2 Gy. In fact, the link between unrecognized/unrepaired DSB and cell survival is now very clearly documented [20]. Literature data converge to the conclusion that one to two unrepaired DSB is a lethal event for human fibroblasts [31], and the same link is true about two to three unrepaired DSB for human tumors since tumors may tolerate more unrepaired DSB than healthy tissues [14,20,32]. Furthermore, we have already shown that HRS phenomenon results in the lack of recognition and repair of DSB by the NHEJ pathway [19,20]. Hence, in order to consolidate the relevance of our data with regard to these two features, we have plotted the number of residual γH2AX foci with the corresponding cell survival data at 2 Gy (SF2) and 0.2 Gy (SF0.2) (Figure 4).

The relationship between the SF2 data, shown in Figure 2, and the corresponding γH2AX foci assessed at 24 h post-irradiation, shown in Figure 3, was found to be in agreement with the link between SF2 and the number of unrepaired DSB reflected by the γH2AX foci described in [32] (Figure 4A). It is also noteworthy that data obtained with a dose of 0.2 Gy obeyed the same relationship (Figure 4A), supporting, again, that a very small number of unrepaired DSB can lead to cellular death. Lastly, the clonogenic cell survival at 0.2 Gy appeared to be inversely proportional to the percentage of DSB recognized and repaired by the NHEJ (Figure 4B). The formula deduced from the data fits suggested that cell survival decreased by 1% each time the percentage of DSB recognized and repaired by NHEJ decreased by one-third (Figure 4B).

### 3.3. Molecular Response to SBRT of the 6 Tumor Cell Lines Used in This Study

In order to further examine the potential impact of the HRS status in the response to SBRT, we applied 2 and 0.2 Gy delivered at one time, 2 Gy delivered in 10 subfractions of 0.2 Gy, and 2 Gy delivered in CyberKnife conditions, with minibeams to the 6 tumor cell lines. The first series was performed with a CyberKnife irradiator of CLB (Lyon, France) with the HT29, A549, and U373 cells. The second parallel series was performed in the same conditions with a CyberKnife irradiator of the CHUV (Lausanne, France).

The γH2AX foci kinetics after the “2 Gy” condition were not found to be significantly different from those obtained with an orthovoltage X-ray irradiator (*p* > 0.2) (Figure 4). After the “0.2 Gy” condition, in the two HRS-positive cells, the number of γH2AX foci was found to be lower at 10 min and higher at 24 h post-irradiation than in the HRS-negative cells (*p* < 0.01) (Figure 5). Interestingly, the “10 × 0.2 Gy” condition resulted in a number of early γH2AX foci, assessed 10 min post-irradiation, similar to that assessed in the “2 Gy” condition (*p* > 0.4) (Figure 5). At 24 h post-irradiation, the “10 × 0.2 Gy” condition resulted in increasing the amount of residual γH2AX foci reflecting unrepaired DSB in the two HRS-positive cells by comparison with the “2 Gy” condition (*p* < 0.02). Conversely, the “10 × 0.2 Gy” condition did not significantly change the number of residual γH2AX foci in the HRS-negative cells (*p* > 0.2). These findings suggest that the severity of DSB only increases in HRS-positive cells (Figure 5). The “2 Gy Cyber” condition was found to amplify the differences already noticed with the “0.2 Gy” condition (*p* < 0.02) again. The “2 Gy Cyber” and “10 × 0.2 Gy” conditions showed the most severe DSB and the largest differences between the HRS-negative and positive cells, (Figure 5). Similar conclusions were reached with the U87, LN229, and SNU475 cell lines.

During our experiments, we observed some cells showing a number of γH2AX foci higher than 100, i.e., they were difficult to score with precision. Such highly damaged cells (HDC) appeared to be a specific feature of the HRS-positive cells that was mainly observed in “10 × 0.2 Gy” and in “2 Gy Cyber” conditions (Figure 6). In order to account for HDC, we have re-expressed the previous data by considering four categories of cells: no γH2AX foci, no more than 15 γH2AX foci, more than 15 γH2AX foci, and HDC. The 24 h data did not show any HDC after a single dose of 2 Gy (Figure 6 and Figure 7). The HDC were observed in HRS-positive cells in the “0.2 Gy” conditions, and such features increased in “10 × 0.2 Gy” and “2 Gy Cyber” conditions in the HRS-positive cells only, suggesting a specific biological dose-enhancement. Conversely, with regards to the HRS-negative cells, a non-significant amount of HDC was observed. No cell with more than 15 foci was observed in HRS-negative cells. (Figure 6 and Figure 7). These findings suggest that a considerable amount of additional DSB may be specifically induced in the HRS-positive cells when the dose is delivered in the hyper-fractionated SBRT conditions.

### 3.4. Molecular Response to SBRT of the 13HNG and AG1521 Skin Fibroblast Cell Lines

The HRS phenomenon is not limited to the tumor cell lines; it has also been observed in healthy tissues such as skin fibroblasts [33,34,35,36,37]. Hence, if the SBRT irradiation would concern the healthy tissues around the tumor, some over-reaction would also be expected. Hence, in order to examine the influence of the HRS status in the response of skin fibroblasts to SBRT, we applied the same irradiation conditions as described above to two untransformed skin fibroblast cell lines: AG1521 (HRS-positive) and 13HNG (HRS-negative) (Figure 8). The HDC were only observed in HRS-positive cells in the “0.2 Gy” and “2 Gy” conditions, and such features increased in “10 × 0.2 Gy” and “2 Gy Cyber” conditions. Such data confirm again that the positive HRS status may lead to a specific biological dose-enhancement in SBRT irradiation conditions (Figure 8).

### 3.5. Quantified Relationship between the HRS/IRR Ratio and the Percentage of HDC

The HDC appeared to be a biomarker specific to a repeated HRS phenomenon. In order to examine whether a quantified link exists between the percentage of HDC and the HRS/IRR ratio, these two parameters, obtained from the four HRS-positive tumor cells, were plotted together (Figure 9). Interestingly, the higher the HRS/IRR ratio, the larger the percentage of HDC. Furthermore, the percentage of HDC increased with the number of SBRT sessions by obeying a linear function close to a one-to-one correlation. Hence, each infinitesimal extent of HRS corresponds to about a 1% increase in HDC. The HRS/IRR ratio is a bounded value, so the percentage of HDC is expected to be a curvilinear function of the number of SBRT sessions. However, further investigations will be needed to document the occurrence of the HRS phenomenon in the SBRT sessions and to verify such a hypothesis.

## 4. Discussion

### 4.1. From Hypofractionated to Stereotactic RT Modalities

The occurrence of the HRS phenomenon in RT depends on the dose, the dose-rate, and the HRS status of the irradiated cells. To date, some hypofractionated and/or stereotactic RT modalities may include conditions in which the HRS phenomenon may occur and may be repeated. In this study, we have applied particular hypofractionated SBRT conditions (those of CyberKnife™ RT). Theoretically, as far as the HRS phenomenon can occur and be repeated, our conclusions may be relevant for any stereotactic delivery of the dose. Furthermore, our results showed that the HRS phenomenon is independent of the tumor cell type (glioblastoma, lung adenocarcinoma…), and therefore, many SBRT or stereotactic radiosurgery (SRS) protocols can be concerned by such conditions. Further investigations are therefore needed to document the occurrence of HRS in other RT modalities.

Historically, hypofractionated RT has long been associated with severe complications, while hyperfractionated RT, with several sessions of about 2 Gy, was known to reduce them [38,39,40]. In the 1930s, the “German School” with Holzknecht and Wintz preferentially applied hypofractionated RT, while the “French school”, represented by the medical staff of the Curie Institute in Paris with Regaud, Coutard, and Del Regato, promoted hyperfractionated RT [38,39,40]. First developed in the 1950s, the stereotactic approach has significantly improved the tumor targeting, which has revived the old debate about hypo or else hyperfractionated RT [41]. Particularly, while hypofractionation may concern other RT modalities, it remains intimately linked to SBRT, and the efficiency of such specific modality has been often attributed, at least partially, to hypofractionation [42,43]. Some authors have suggested that long periods between two RT sessions may facilitate immune reactions, tumor reoxygenation, or repopulation, but these last features have not been correlated to radiosensitivity in a unified model yet [14,44,45]. Furthermore, immune reactions and tumor repopulation require a significant cell killing effect. With regards to DNA damage, if the 24 h repair time is protracted, no significant number of additional DNA damage is created or repaired, and therefore, no additional cell killing effect is produced [14]. In other terms, by considering unrepaired DNA breaks, protracting the time between RT sessions over 24 h should not protect significantly more healthy tissues or kill more tumor cells and, consequently, should not favor the occurrence of immune reactions and tumor repopulation [14]. In a hypofractionated RT modality (whether SBRT or not), hypofractionation of the total dose is likely to produce additive, but not supra-additive, effects. Conversely, by not considering the hypofractionation of high doses delivered during all the treatment but the hyperfractionation of low doses during each session, a significant dose-enhancement may be observed if the HRS phenomenon occurs, i.e., if some doses and dose-rates obey HRS conditions [18]. In addition, by producing an excess of cell lethality, HRS may also impact on vascular damage and immune response. Hence, our model is compatible with other hypotheses evoked above.

### 4.2. HRS Phenomenon and Stereotactic RT

The HRS phenomenon results in a significant reduction in clonogenic cell survival, increase in chromosome breaks, micronuclei, and unrepaired DSB, but also an increase in mutation frequency after a single low-dose ranging between 1 and 800 mGy [18,46]. At a dose-rate higher than 1 Gy/min, HRS generally occurs between 1 and 800 mGy. At 0.1 Gy/min, HRS is expected at few cGy [18]. In 2016, we have provided a relevant biological interpretation of the HRS phenomenon in the frame of the radiation-induced ATM nucleoshuttling (RIANS) model [17]. Briefly, in the RIANS model, IR induce DNA damage (similarly to DSB), but they also induce monomerization of the cytoplasmic ATM dimers in a dose-dependent manner. The resulting ATM monomers diffuse to the nucleus to participate in the DSB recognition by NHEJ [47]. The delay of the RIANS was shown to be correlated with the cell survival and severity grade of post-RT tissue reactions [20,23]. The RIANS is very fast (some minutes after 2 Gy) in radioresistant cells (SF2 > 60%) called group I cells. It is not functional in hyperradiosensitive cells (SF2 < 10%) called group III cells. In cells showing significant but intermediate radiosensitivity (10% < SF2 < 60%), called group II cells, the RIANS is delayed by the cytoplasmic over-expression of some proteins called X-proteins [17,47]. At low doses, the number of ATM monomers that diffuse in the nucleus and the number of DSB induced by IR decrease together but not at the same rate. Indeed, the number of DSB induced by IR decreases by obeying an induction rate of about 40 DSB induced per Gy per cell, regardless of the group of radiosensitivity. In group II cells, because ATM monomers are sequestrated by X-proteins in the cytoplasm, the number of ATM monomers decreases faster than in group I cells. Consequently, the ratio of ATM monomers that diffuse in the nucleus per DSB is much lower in group II than group I cells [17,47]. In a specific mGy dose range, the flux of ATM monomers may not be sufficient for recognizing the few DSB induced. In the case of non-recognition of some DSB, the survival rate, the number of unrepaired DSB, micronuclei, or mutations may be affected significantly. Hence, the HRS-positive cells necessarily belong to the group II, i.e., they are characterized by a moderate radiosensitivity and a delayed RIANS. It must be stressed than such a hypothesis is relevant for both healthy tissues and tumors [17,47].

### 4.3. Evidence That Each SBRT Session Results in a Repetition of Low Doses

Through the graphical demonstration of Figure 1, there is evidence that each SBRT session results in a repetition of low doses. Hence, let us consider the cell survival of the irradiated tumor S(d), at a dose d, delivered by one single minibeam: the resulting cell survival S_SBRT_ after one SBRT session of ρ repeated doses delivered by ρ minibeams is:S_SBRT_ = [S(d)]^ρ^(3)

By using the HT29, A549, and U373 tumor data described above and applying the Equation (3), let us consider that each minibeam delivered 10 cGy on average: the HRS-negative cells show a cell survival S(d) of about 0.9999, while the HRS-positive cells show a cell survival S(d) ranging from 0.98 to 0.93. Hence, by taking a numerical example with ρ = 10 minibeams overlapping in the same region of the tumor, the resulting cell survival S_SBRT_, after one SBRT session for HRS-negative and positive tumors, is 0.9999^10^ = 0.999 and 0.98^10^ = 0.817 to 0.91^10^ = 0.39, respectively. Hence, in the HRS-positive cells exposed to one SBRT session, the final survival rate may reach a survival from 1.2 to 2.5 times lower than the HRS-negative cells. Altogether, such experimental and calculated data are consistent with a significant dose enhancement due to the specific occurrence of an HRS phenomenon repeated several times during each SBRT session.

### 4.4. The HRS Phenomenon in Stereotactic RT Modalities: Towards New Predictive Assays?

If the HRS-positive cells show an enhanced killing effect, our findings raise two important questions: (1) What would be the effect on healthy tissues if they are also HRS-positive? (2) How can we predict whether a tumor/healthy tissue is HRS-positive or negative? For the first question, it appears obvious that healthy tissues can also be HRS-positive and that the HRS phenomenon is not a specific feature of tumors, as demonstrated by the Slonina’s group [33,34,35,36,37]. However, it is important to note that the HRS statuses of tumor and healthy tissues, as with their radiosensitivity at high doses, are independent. If healthy tissues are HRS-positive, or even if both healthy tissues and tumors are HRS-positive, the benefit of a hyperfractionation of low-doses may be reduced. However, it must be stressed again that, in the particular case of SBRT, the tumor targeting is particularly precise, and therefore, healthy tissues may be spared more than with any other RT modality. The benefit of SBRT may be not negligible in this case.

With regards to the second question, the frequency of a positive HRS status in tumors is still unknown. However, a predictive assay, based on the irradiation of 2 and 0.2 Gy at high dose rate by taking DSB biomarkers as endpoints, may be informative enough to verify the HRS status of a given tumor, even if the feasibility of such a specific assay may be strongly linked to the accessibility of the tumor to perform biopsies. Fortunately, in radiation oncology, a tumor sample is, in the vast majority of cases, a prerequisite before irradiation, whether in the form of a surgical specimen or a biopsy. With regard to healthy tissues, it is always possible to perform cutaneous biopsies or blood lymphocyte assays to verify the HRS status.

### 4.5. Towards a Biology-Guided Radiation Therapy (BGRT)?

In the long term, determining the HRS status of both tumors and healthy tissues, may improve RT efficiency and increase its safety. Indeed, healthy tissues exhibiting an HRS-negative radiosensitivity surrounding an HRS-positive tumor would be a good candidate for a treatment technique favoring discontinuous microbeams, such as CyberKnife™, rather than volumetric modulated arc therapy (VMAT) or 3D RT modalities that involve a few beams. In this case, a de-escalation of the dose could even be considered in order to reduce the risk of radiotoxicity while safely maintaining anti-tumor efficiency. It must be stressed that such a biomolecular selection of the patients would be entirely compatible with all current SBRT techniques, allowing for increased treatment precision [48,49].

## 5. Conclusions

As mentioned in the above paragraph, the practical requirements for the occurrence of the HRS phenomenon depends on some specific ranges of dose and dose-rate. For example, some 3D-conformational RT protocols (i.e., several non-coplanar arcs) may also include such conditions and concern HRS-positive cells. Hence, our approach should be considered as a first step of investigations about the impact of the HRS phenomenon on certain modalities of RT. For the first time to our knowledge, it is suggested, here, that the benefit of the SBRT modalities may not be due to the hypofractionation of the dose through the whole treatment but, rather, to the hyperfractionation of the low dose delivered by each minibeam during each session, as well as the occurrence of the HRS phenomenon, in cells exhibiting HRS positive status. Further experiments are needed to document these findings in other SBRT modalities and, even, to better evaluate the risk of exposing HRS-positive healthy tissues and the benefit of treating HRS-positive tumors.

## Figures and Tables

**Figure 1 cancers-15-03979-f001:**
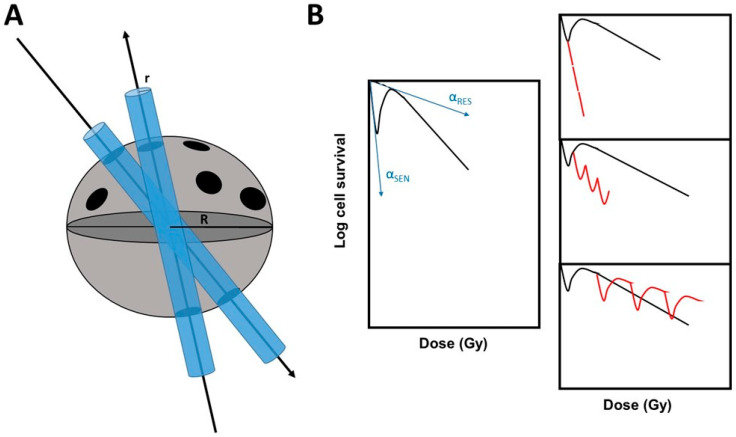
SBRT and the hypersensitivity to low doses phenomenon (HRS). (**A**) A schematic view of SBRT centripetal minibeams crossing a spherical tumor. (**B**) On the left, there is a representative example of a survival curve showing HRS phenomenon. The HRS parameters defined in Material and Methods are indicated. On the right, the survival effect of repeated low doses in an HRS-positive cell at different doses is shown. Red lines indicate the expected fractionation effect.

**Figure 2 cancers-15-03979-f002:**
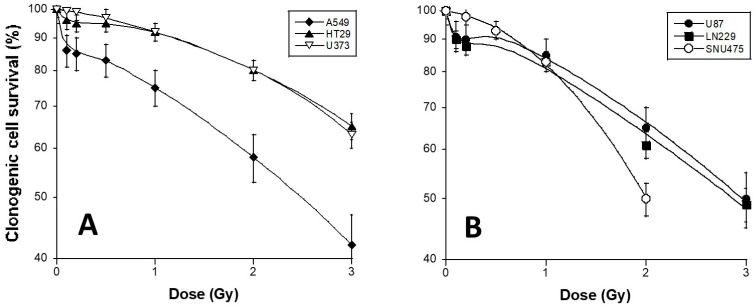
Clonogenic cell survival of the 6 tumor cell lines used in this study. Cell survival data were plotted as a function of X-ray dose delivered at high dose rate (4 Gy/min) by a standard orthovoltage irradiator. Each point represents the mean of triplicate experiments ± SEM. Survival data were fitted to the LQ model and HRS LQ variant model. The HRS-positive cell lines are represented by closed symbols, while the HRS-negative cell lines are represented by open symbols. The (**A**) panel represents the A549, HT29, and U373 data and the (**B**) panel represents the U87, LN229, and SNU475 data.

**Figure 3 cancers-15-03979-f003:**
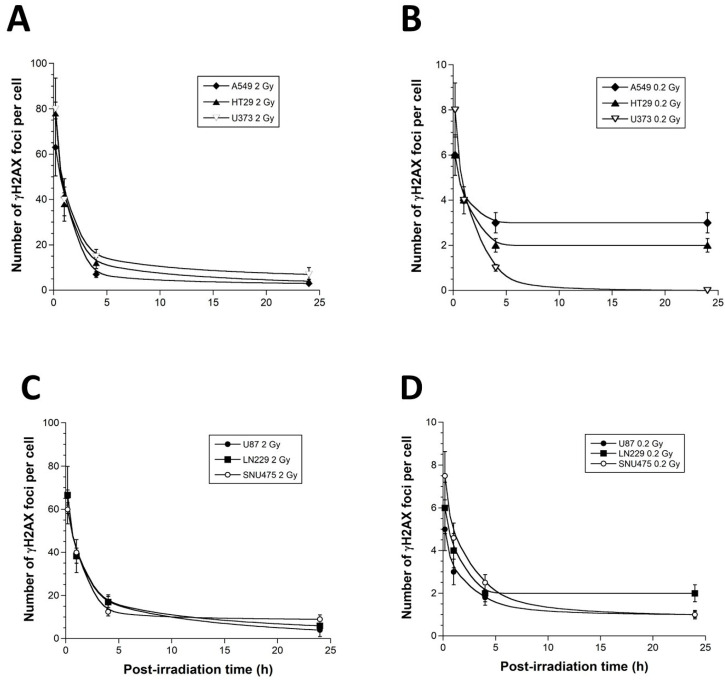
Radiation-induced γH2AX foci kinetics of the 6 tumor cell lines used in this study. The γH2AX foci data were expressed as a number of γH2AX foci per cell, as a function of post-irradiation time, after a dose of 2 Gy (**A**,**C**) and 0.2 Gy (**B**,**D**) X-ray delivered at high dose rate (4 Gy/min) by a standard orthovoltage irradiator. Each point represents the mean of triplicate experiments ± SEM.

**Figure 4 cancers-15-03979-f004:**
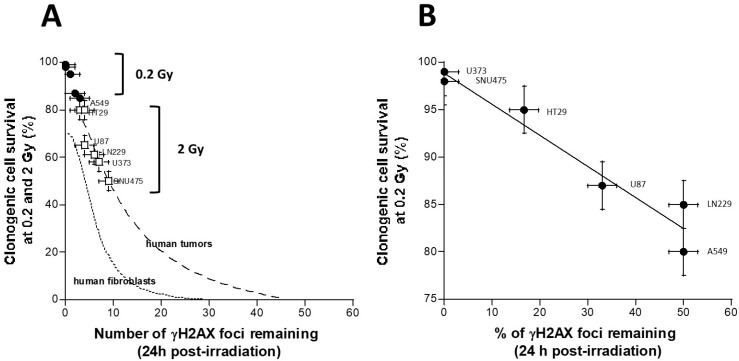
Relationships between the cell survival and the γH2AX data. A. The survival data assessed at 0.2 (SF0.2) and 2 Gy (SF2), shown in Figure 2, were plotted against the corresponding γH2AX data assessed at 24 h post-irradiation, shown in Figure 3, and obtained from the indicated human tumor cell lines. Each point represents the mean of triplicate experiments ± SEM. The dashed and the dotted lines represent the links between the SF2 and the number of residual γH2AX foci obtained from 40 human tumor cell lines described in [32], as well as 200 human fibroblast cell lines described in [20]. (**B**) The survival data assessed at 0.2 (SF0.2), shown in panel (**A**), were plotted against the corresponding percentage of γH2AX data assessed at 24 h post-irradiation (24 h data divided by 10 min data, shown in Figure 3, and obtained from the indicated human tumor cell lines). Each point represents the mean of triplicate experiments ± SEM. The line represents the formula y = 98.8 − 0.32x (r = 0.97).

**Figure 5 cancers-15-03979-f005:**
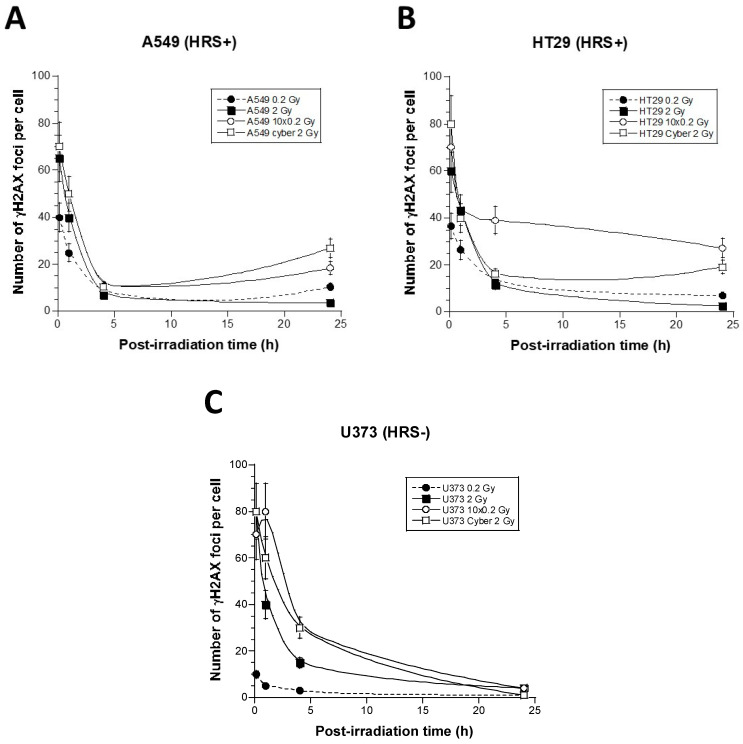
Radiation-induced γH2AX foci kinetics of the A549 (HRS-positive) (**A**), HT29 (HRS-positive) (**B**), and U373 (HRS-negative) (**C**) tumor cell lines irradiated with different SBRT delivery mode. For each cell line, the γH2AX foci data were expressed as a number of γH2AX foci per cell and a function of post-irradiation time (top) after a single dose of 2 Gy (2 Gy), 0.2 Gy (0.2 Gy), a series of 10 times 0.2 Gy (10 × 0.2 Gy), and a dose of 2 Gy delivered by CyberKnife (2 Gy Cyber). Each point represents the mean of triplicate experiments ± SEM.

**Figure 6 cancers-15-03979-f006:**
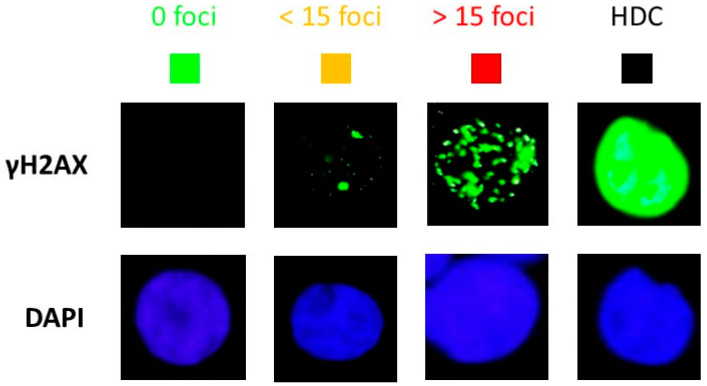
Representative images of the four categories of cells observed 24 h post-irradiation: cells showing no γH2AX foci, no more than 15 γH2AX foci, more than 15 γH2AX foci, and HDC. The γH2AX staining is in green, and the corresponding DAPI counterstaining is in blue.

**Figure 7 cancers-15-03979-f007:**
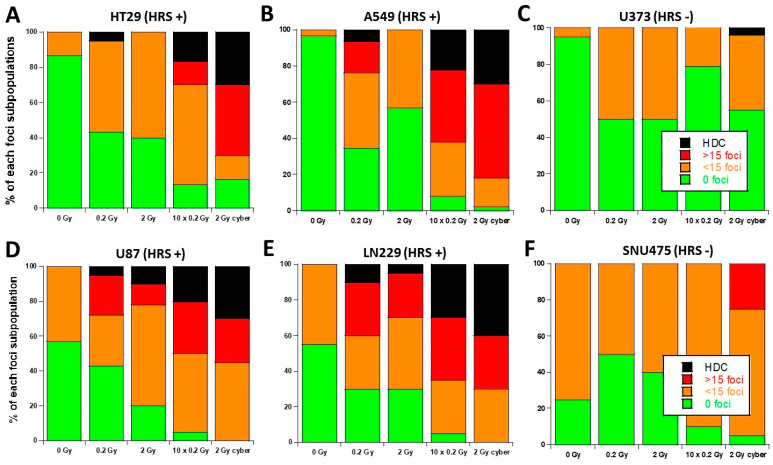
Stacked bar chart of 24 h γH2AX foci data from the 6 tumor cell lines used this study. The 24 h data obtained from the 6 indicated tumor cell lines were re-expressed by considering four categories of cells defined in Figure 6. Each point represents the mean of triplicate experiments ± SEM.

**Figure 8 cancers-15-03979-f008:**
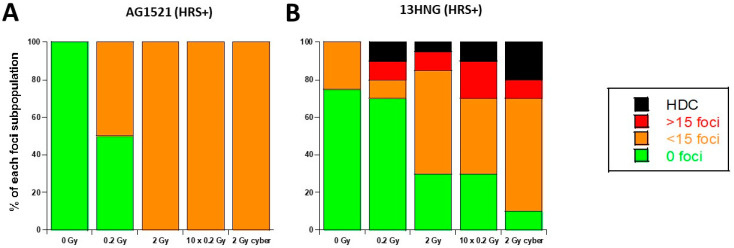
Stacked bar chart of 24 h γH2AX foci data. The 24 h data obtained from the 2 indicated heathy tissue cell lines were re-expressed by considering four categories of cells defined in Figure 6. Each point represents the mean of triplicate experiments ± SEM.

**Figure 9 cancers-15-03979-f009:**
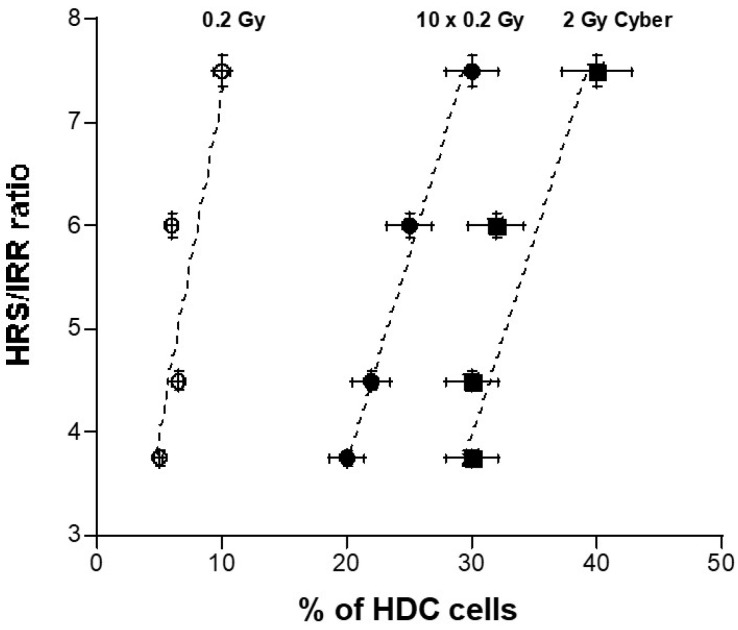
HRS/IRR ratio as a function of percentage of HDC. The HRS/IRR ratios mentioned for the four HRS-positive tumor cells in Table 2 were plotted against the corresponding percentage of HDC in the 0.2 Gy, 10 × 0.2 Gy, and 2 Gy Cyber conditions shown in Figure 6. Dotted lines correspond to a linear function used for data fitting.

**Table 1 cancers-15-03979-t001:** Biological features of the tumor and fibroblast cell lines used in this study.

Cell Line	Origin		HRS Status
HT29	Grade II colon adenocarcinoma isolated from a primary tumor from a white Caucasian female patient	#HTB-38, ATCC ^a^	+
A549	lung carcinoma from a white Caucasian male patient	#CCL-185, ATCC ^a^	+
U373 MG	glioblastoma astrocytoma from a malignant tumor	#08061901, ECACC ^b^	−
U87 MG	from malignant glioma from a male patient	#HTB-14, ATCC ^a^	+
LN229	from the right frontal parieto-occipital cortex of a white female patient with glioblastoma.	#CRL-2611, ATCC ^a^	+
SNU475	from an hepatocellular carcinoma	#CRL-2236 ATCC ^a^	−
AG1521	Skin (foreskin) fibroblast from a foreskin of a 3 day child	#AG01521, Coriell Institute Repository ^c^	−
13HNG	Skin fibroblast from a cancer patient showing adverse tissue after radiotherapy	COPERNIC Collection	+

^a^ American Type Culture Collection (ATCC)*, (Manassas, VA, USA); ^b^* European Collection of Authenticated Cell Cultures (ECACC), Health Security Agency, (Salisbury, UK); ^c^ Coriell Institute Repository (Camden, NJ, USA). + and – mean HRS-positive and -negative status, respectively.

**Table 2 cancers-15-03979-t002:** Numerical values of the adjustable parameters defining the variant LQ model for the tumor cell lines used here *.

Cell Lines	HRS Status	α_res_ (Gy^−1^)	g	Dc (Gy)	HRS/IRR Ratio	β (Gy^−2^)	SF2 (%)
U373	−	0.05	0	NA	1	0.029	58 ± 6
SNU475	−	0.05	0	NA	1	0.14	50 ± 4
A549	+	0.273	8.11	0.17	4.5	0.0003	80 ± 4
HT29	+	0.001	12.63	0.15	6	0.003	80 ± 6
U87	+	0.14	12.05	0.13	3.75	0.03	65 ± 6
LN229	+	0.19	7.03	0.15	7.5	0.016	61 ± 3

* Survival data shown in Figure 2 were fitted to S(D) = exp(−α_res_ (1 + g exp(−D/Dc))D − βD^2^) (Formula (2)). It is noteworthy that all the data fit provided correlation coefficients larger than 0.98; + and – mean HRS-positive and -negative status, respectively; NA: non applicable.

## Data Availability

The data presented here are either present in a deposed database (see Section 2) or will be made available on reasonable request.

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
