# Peer review of "Influence of the Hypersensitivity to Low Dose Phenomenon on the Tumor Response to Hypofractionated Stereotactic Body Radiation Therapy"

_cancers, 2023, doi:10.3390/cancers15153979_

Round 1
Reviewer 1 Report
The manuscript entitled “Influence of the hypersensitivity to low dose phenomenon on the tumor response to hypofractionated stereotactic body radiation therapy” by Le Reun et al. aimed to the hypersensitivity to low dose in different tumor cells, evaluating the influence of HRS status on the response of healthy tissues to SBRT. The manuscript deals with an interesting topic which could enrich the current literature. Additionally, considering the increasing technological advance in several fields, as well the radiation therapy, the continuous testing of hypotheses is fundamental to provide a growing piece of evidence for further research. A few corrections are suggested in order to improve the quality of a manuscript which is already more than presentable.
INTRODUCTION
87-97: write also a sentence which clearly states the aim of your study.
MATERIALS AND METHODS
167: albeit is already described, briefly report this description.
RESULTS
Figure 2: improve the quality of the figure in terms of resolution.
Figure 3-4: as before.
280: Avoid personal considerations in this section. Only report the data. Any further comments or considerations (which are well welcomed in a successive interpretation of data) could be moved to the discussion section.
Figure 5: considering the great number of figures fused into one, you could divide and increase the resolution of the figure. Additionally, the paragraph is too long.
DISCUSSION
The discussion is well made however there are two things lacking.
The first is a paragraph regarding future perspectives and the potential role of your findings in a common clinical practice or further research. To this regard, I suggest you to also briefly recall how SBRT could be now used in several clinical scenarios. See also: DOI: 10.3390/diagnostics12081981, doi: 10.21037/jtd.2016.03.14
The second is the lack of limitations which should be reported in any case.
minor typos
Author Response
Replies to the reviewer 1 ‘s comment
We thank the reviewer for his/her comments
The manuscript entitled “Influence of the hypersensitivity to low dose phenomenon on the tumor response to hypofractionated stereotactic body radiation therapy” by Le Reun et al. aimed to the hypersensitivity to low dose in different tumor cells, evaluating the influence of HRS status on the response of healthy tissues to SBRT. The manuscript deals with an interesting topic which could enrich the current literature. Additionally, considering the increasing technological advance in several fields, as well the radiation therapy, the continuous testing of hypotheses is fundamental to provide a growing piece of evidence for further research. A few corrections are suggested in order to improve the quality of a manuscript which is already more than presentable.
INTRODUCTION
87-97: write also a sentence which clearly states the aim of your study.
OK See modified text
MATERIALS AND METHODS
167: albeit is already described, briefly report this description.
OK See modified text
RESULTS
Figure 2: improve the quality of the figure in terms of resolution.
Figure 3-4: as before.
OK See modified figures
280: Avoid personal considerations in this section. Only report the data. Any further comments or considerations (which are well welcomed in a successive interpretation of data) could be moved to the discussion section.
OK See modified text
Figure 5: considering the great number of figures fused into one, you could divide and increase the resolution of the figure. Additionally, the paragraph is too long.
OK see modified figure. We have deleted the figures showing the data expressed as the percentage of foci remaining and the paragraphs were shortened.
DISCUSSION
The discussion is well made however there are two things lacking.
The first is a paragraph regarding future perspectives and the potential role of your findings in a common clinical practice or further research. To this regard, I suggest you to also briefly recall how SBRT could be now used in several clinical scenarios. See also: DOI: 10.3390/diagnostics12081981, doi: 10.21037/jtd.2016.03.14
The second is the lack of limitations which should be reported in any case.
OK See the additional paragraph. See modified text.
Reviewer 2 Report
The Authors investigated the effect of low dose hypersensitivity on tumor response in cell lines given hypofractionated SBRT. The topic is of interest. The manuscript is clear and well written. Methodology is robust. The whole study is scientifically sound. Few comments hereby:
Introduction
1) Can you please define stereotactic radiotherapy (dose per fraction, number of fractions, dose gradient, ablative purpose et al?)
2) Does this study apply also to stereotactic radiosurgery (SRS)? Or only extracranial localizations (SBRT)?
3) Can you please least the mechanisms by which SBRT exploits its effect? (DNA damage, vascular damage, immunogenic cell death)
4) Does the theory of non-coplanar minibeams applies also to volumetric techniques?
Moderate revision required
Author Response
Replies to the reviewer 2 ‘s comment
We thank the reviewer for his/her comments
The Authors investigated the effect of low dose hypersensitivity on tumor response in cell lines given hypofractionated SBRT. The topic is of interest. The manuscript is clear and well written. Methodology is robust. The whole study is scientifically sound. Few comments hereby:
Introduction
1) Can you please define stereotactic radiotherapy (dose per fraction, number of fractions, dose gradient, ablative purpose et al?)
See modified text in Introduction
2) Does this study apply also to stereotactic radiosurgery (SRS)? Or only extracranial localizations (SBRT)?
Our conclusions may be relevant for any stereotactic delivery of the dose, i.e. when the dose is dispatched and delivered in several microbeams, each delivering low-dose
Our results showed that the phenomenon is independent from the tumor cell type (glioblastoma, lung adenocarcinoma…), and hence the localization (SBRT or SRS).
Regarding the CyberKnife, one fraction delivers about 95-200 microbeams (cf. Lipani J. Tumors of the Spine 2008, Chapter 16 - Stereotactic Radiosurgery of the Spine. Pages 346-355. 2008); then:
- considering a low total dose per fraction such as 7 Gy (eg. bone met SBRT), one microbeam could deliver at least 035 Gy(7/200), and up to 0.07 Gy (7/95), which are both doses able to elicit HRS phenomenon
- considering a high total dose per fraction such 24 Gy (eg. brain met SRS), one microbeam could deliver at least 12 Gy(24/200), and up to 0.25 Gy (24/95), which are both doses able to elicit HRS phenomenon
See modified text in Discussion in Section 4.1
3) Can you please least the mechanisms by which SBRT exploits its effect? (DNA damage, vascular damage, immunogenic cell death)
We have already evoked the different molecular interpretations of hypofractionation and SBRT available in literature (see 4.1 from line 399). However, HRS phenomenon as interpreted by RIANS model is due to a delayed radiation-induced ATM nucleoshuttling. However, the consequences of HRS may also concern vascular damage and immunogenic cell death linked to the debris caused by the excess of cell death. See modified text line 416.
4) Does the theory of non-coplanar minibeams applies also to volumetric techniques?
- The notion of dose repetition is here crucial, and particularly the time between to microbeams:
- If two microbeams are sufficiently spaced out in time, both would be able to successively elicit an HRS phenomenon è this is the case of step and shoot modalities: i) CyberKnife, ii) and in a lesser extent, IMRT and 3D with a lot of beams (not very common in routine)
- If two microbeams are delivered too close together in time, their dose would add up to a higher dose that will be beyond HRS domain è this is the case of rotational IMRT (VMAT, HyperArc, and Tomotherapy)
See modified text in the new section 4.5
Round 2
Reviewer 1 Report
The authors improved the manuscript accordingly to previous suggestions.
minor typos
Reviewer 2 Report
I do not have any further comment
Acceptable quality of english language